# Segmental Assessment of Trunk Control in Moderate-to-Late Preterm Infants Related to Sitting Development

**DOI:** 10.3390/children8090722

**Published:** 2021-08-24

**Authors:** Noppharath Sangkarit, Orawan Keeratisiroj, Ponlapat Yonglitthipagon, Surussawadi Bennett, Wantana Siritaratiwat

**Affiliations:** 1Research Center in Back, Neck, Other Joint Pain and Human Performance (BNOJPH), School of Physical Therapy, Faculty of Associated Medical Sciences, Khon Kaen University, Muang, Khon Kaen 40002, Thailand; noppharath.sa@kkumail.com (N.S.); surmac@kku.ac.th (S.B.); 2Faculty of Public Health, Naresuan University, Phitsanulok 65000, Thailand; orawansa.nu@gmail.com; 3School of Physical Therapy, Faculty of Associated Medical Sciences, Khon Kaen University, Khon Kaen 40002, Thailand; ponlapat@kku.ac.th

**Keywords:** postural control, sitting development, premature home-raised infant, Alberta Infant Motor Scale, Segmental Assessment of Trunk Control

## Abstract

(1) Background: The assessment of postural segment control in premature infants seems to be critical during the onset of upright gross motor development, especially sitting. Identifying correlations between postural segment control and the development of sitting milestones could help with promoting optimal gross motor movement. However, data on this topic in home-raised premature infants via longitudinal design are still limited. The purpose of this study was to examine relationships between postural segment control and sitting development through series assessments from the corrected age of 4 months until the early onset of independent sitting attainment. (2) Methods: 33 moderate-to-late premature infants were recruited. Their trunk segment control was assessed using the Segmental Assessment of Trunk Control (SATCo), and sitting development was examined by the Alberta Infant Motor Scale (AIMS). Relationships between SATCo and sitting scores were analysed using Spearman’s rank correlation (r_s_). (3) Results: significant fair-to-good correlations between segmental trunk control and sitting scales were found from 4 months (r_s_ = 0.370–0.420, *p* < 0.05) to the age of independent sitting attainment (r_s_ = 0.561–0.602, *p* < 0.01). (4) Conclusion: relationships between the trunk segment control and sitting ability of moderate-to-late preterm infants were increased in accordance with age.

## 1. Introduction

The sitting skill is an essential development and consequently affects the subsequent fundamental motor development during childhood. A World Health Organisation multicentre study reported that typically developing infants show a range of ability to sit without support in the age from 3.8 to 9.2 months [1]. Karasik et al. showed that the early age for sitting attainment in full-term infants was approximately 5 months, depending on their different sitting practises and experiences [2]. The ability to control sitting balance gradually emerges in typically developing infants during the period from 2 to 9 months.

Premature infants may appear to be delayed sitters compared with full-term infants. A previous study reported that 50% of 37 moderate-to-late premature infants displayed delayed unsupported sitting at 10 months of age, while more than 9% had delayed unsupported sitting at 13 months of age [3]. In addition, 93 infants born between 24 and 35 weeks of gestation showed delayed sitting development at the corrected ages of 6 and 12 months [4].

Postural control works across multiple trunk segments during the typical development of sitting. The trunk control of full-term and premature infants develops in a cephalocaudal pattern of upright control [5,6,7]. These studies confirmed that trunk-control development occurs by the contribution of unique challenges created by many muscles and skeletal subunits. As trunk control is formed by many different anatomical regions, trunk control or postural balance in sitting should be evaluated in segments [5]. Butler et al. (2010) examined the psychometric properties of the Segmental Assessment of Trunk Control (SATCo) and found that the SATCo is a valid and reliable assessment of trunk control in segmental ways [8]. Segmental trunk control is generally considered to be evidence for a deviation of the acquisition of gross motor development in preterm infants. Sato and Tudella reported that only 10% of 20 late premature infants achieved full trunk control in sitting at ages of 6 to 8 months old, while 70% of 36 typically developing full-term infants achieved full trunk control at 8 months of age [5].

Previous studies segmentally demonstrated an association between gross motor development, especially sitting milestones, and trunk control [7,9,10,11,12]. Duncan et al. found that proficiency at gross motor skills was influenced by the varying levels of parent holding of the infants’ trunk in sitting and standing positions of full-term infants aged 1 to 8 months [7]. A cross-sectional study by Greco et al. [12] verified segmental postural control during the emergence of gross motor performance in prone, supine, sitting, and standing positions at 6 and 7 months of age in full-term infants and those with 33 weeks gestational age and having no comorbidity from prematurity. A current study indicated that achieving trunk control at the midthoracic level of full-term infants was linked to the ability to sit independently at 6 months of age, whereas this association appears in the lower thoracic level of the segmental trunk in preterm birth infants at 7 months corrected age [12].

A longitudinal study by Pin et al. examined the development of segmental trunk control in 31 very preterm infants, with a mean gestational age of 27.2 weeks, and 30 full-term infants from 4 to 12 months. This previous study confirmed that the very preterm infants generally had a delay in developing the same level of segmental trunk control in sitting at all age levels, compared with that of their full-term peers [10]. Interestingly, the same group showed a nonsignificant correlation between SATCo and AIMS scores at 4 months in both groups of infants [11]. Furthermore, two studies found a correlation between segmental trunk control and the different postures of gross motor development in supine, sitting, and standing positions at different ages in preterm and full-term infants [11,12]. A high capacity to segmentally control the trunk was associated with a higher level of gross motor performance [11]. There were limited longitudinal data on the relationship between the development of trunk segment control and sitting ability [12]. We hypothesised there would be relationships between segmental trunk control and sitting ability from the age of learning to sit or at 4 months corrected age onwards. 

The purpose of this study was to examine relationships between postural segment control and sitting development through series assessments from the corrected age of 4 months until the early onset of independent sitting attainment. The series assessments in this study examined segmental trunk control within three conditions—static, active, and reactive control—and the results of the study would also display how late preterm infants progress with their segmental postural control during the sitting milestones from 4 months corrected age until the early onset of independent sitting. 

## 2. Materials and Methods

### 2.1. Participants

Infants were recruited from a name list of infants with premature birth that had been obtained from 26-district health-promoting hospitals within a range of 50 km from Khon Kaen University. The simple sampling of every second name of infants was used to recruit a representative sample from the name list. Moderate-to-late premature infants born between 32 and 36 weeks of gestation were recruited at 4 months corrected age for the first data collection if infants had a stable health condition. Exclusion criteria were seizure, visual, or hearing impairment, congenital abnormality, major brain damage, periventricular leukomalacia (PVL) more than Grade I [13], intraventricular haemorrhage (IVH) [14] more than grade II, or NICU stay of more than 17 days.

The sample size for this study was calculated using the values of correlation coefficient (r = 0.6) between the level of segmental trunk control and the AIMS sitting subscale scores from the previous study conducted in 31 preterms (mean gestation, 27 weeks with low birth weight) and 30 full-term infants aged from 4 to 12 months [11]. The significance level was set at an alpha lower than 0.05 (Z_α⁄2_ = 1.96). The minimal number of required participants was 21. An extra 20% of 21 participants were included in anticipation of any dropouts [11]. In total, 33 infants were considered an adequate sample size for statistical analyses in this prospective analytical study.

### 2.2. Measurement Instruments

#### 2.2.1. Segmental Assessment of Trunk Control (SATCo)

The SATCo is a clinically valid measure for the examination of trunk control at various levels of support in children [8]. The SATCo assesses the level of trunk control in the cephalocaudal direction, altered by manually changing the level of trunk support from the shoulder girdle down to a pelvic segment. This includes 7 parts of trunk segments consisting of the head, upper thoracic, middle thoracic, lower thoracic, upper lumbar, lower lumbar, and full trunk controls (Figure 1). The level of functional segment control of the SATCo was indicated by numbers: 1 = head, 2 = upper thoracic, 3 = midthoracic, 4 = lower thoracic, 5 = upper lumbar, 6 = lower lumbar, and 7 = full trunk control [8]. The assessment of each segment is performed under 3 conditions—static, active, and reactive control [8,15]. Static control is credited when infants maintain a neutral posture at different levels of support. Active control is the ability to maintain the body in a neutral posture while the infant is stimulated to turn the head to either side and return back to the midline. Reactive control is counted when infants can maintain a neutral posture during an external perturbation. The presence or absence of control is recorded as 1 or 0, respectively. ‘NT’ is given when infants are not ready for the test. The reactive condition is not tested at the head segment. 

#### 2.2.2. Alberta Infant Motor Scale (AIMS)

The AIMS is a standardised observational instrument designed and validated to measure changes in gross motor development over time in infants aged 0 to 18 months [16]. The AIMS consists of 21 items in a prone, 9 items in a supine, 12 items in a sitting, and 16 items in a standing position. Movement can be inspired by toys. In order to observe spontaneous movements from the infants, they are minimally touched during an assessment. Except for infants who cannot change position to sitting and standing, the assessor can place them in those postures. Each item is scored as ‘observed’ or ‘not observed’. The lowest observed item and the highest observed item in each position build a window of motor development. The total point values for items before the window are credited as ‘previous items’. Subscale scores are from observed items in the window combined with the previous items. All subscale scores are summed up as the total scores. Testing time takes approximately 15 min for each infant. This study used the AIMS Thai version [17], which revealed high inter- (ICC = 0.988, 95% CI = 0.976–0.994) and intra- (ICC = 0.995, 95% CI = 0.989–0.998) rater reliability.

### 2.3. Ethical Clearance 

The ethical approval to perform this study was obtained by the Khon Kaen University Ethics Committee for Human Research on the basis of the Declaration of Helsinki and the ICH Good Clinical Practice Guideline (Institutional Review Board Number: IRB00008614, protocol ID no.: HE622153, 10 July 2019). The research protocol of this study was renewed from the Khon Kaen University Ethics Committee for Human Research on the basis of the Declaration of Helsinki and the ICH Good Clinical Practice Guideline (Institutional Review Board Number: IRB00001189, 30 June 2020). Data collection was performed from July 2019 to February 2020. The researcher made appointments to collect the data after the parents had given their informed consent to participate voluntarily and allowed their infants to be assessed for trunk control while sitting and gross motor ability. The data were anonymised in order to not reveal patients’ identities, and analysis was conducted in a way in which the final results could not be linked to individual patients.

### 2.4. Procedure

Demographic data of preterm infants were recorded from the personal health booklet, including birth weight, birth height, head circumference, Apgar score at 5 min, gestational age, age at admission (days), and gender. Two paediatric physical therapists with 3 years of clinical experience performed the reliability tests for each measurement. The raters practised performing the tests for 6 months before data collection. Intrarater reliability of the SATCo was performed by the first paediatric physical therapist with a one-month interval using the randomised video recordings of 25 preterm infants aged four, six, and nine months. These infants in the reliability tests were not included in the participants of the main study.

Postural control during sitting was assessed by the first paediatric physical therapist (N.S.) using the SATCo, and gross motor development was examined by another paediatric physical therapist (R.T.) using the AIMS Thai version. Two assessors performed the tests independently, and the scores of the two separate examiners were blinded to each other. Both tests were performed on a monthly basis, starting at the corrected age of 4 months until infants had attained independent sitting. The onset of independent sitting in this study is defined as the ability of infants to sit up straight with the head erect, without hands to balance or support the position, momentarily for at least 10 s [18]. Subsequent assessments occurred on the same date (plus or minus 5 days) of every month.

### 2.5. Data Analysis

Demographic data of participants were reported using descriptive statistics. Due to the normality of the data, the SATCo scores of all conditions and the level of segmental trunk control were reported as median and range, while each subscale and total AIMS scores were reported as mean and standard deviation (SD). The correlations between the segmental trunk control at each condition and the sitting subscale of the AIMS were analysed using Spearman’s rank correlation (R_s_). A rho of 0.25 was considered to be little or no correlation, 0.25 to 0.50 = fair, 0.50 to 0.75 = moderate-to-good, and more than 0.75 was good-to-excellent correlation [19]. All data analyses were performed using SPSS for Windows version 17.0 (licensed by Khon Kaen University, Khon Kaen, Thailand).

## 3. Results

The intrarater reliability (ICC (3,1)) of the SATCo was reported to be 0.936 (95% CI 0.860–0.971). The intrarater reliability (ICC (3,1)) of the AIMS, completed by the second paediatric physical therapist in 25 infants aged from 1 to 12 months, was 0.979 (95% CI, 0.951–0.991).

Of the infants, 33 had a mean age of first assessment at 4 months and 5 days corrected age, and 67% were male. There were neonatal complications—27 had low birth weight, 9 infants had a history of mild bronchopulmonary dysplasia, 5 infants had a history of jaundice, and 1 infant was with G6PD deficiency. The mean (SD) age of their mothers was 27 (7) years old.

Table 1 presents the characteristics of all the preterm infants at inclusion. The mean age of attaining independent sitting for the infants in this study was 7 months, 9 days (SD, 1 month, 8 days) corrected age. Of these infants, 9% exhibited independent sitting at 6 months, 55 % at 7 months, 27% at 8 months, and 9% at 9 months corrected age. 

Table 2 shows the median and range of the SATCo scores in three conditions at the corrected age of 4 and 5 months, and at the age of attaining independent sitting. The median of the SATCo scores in each condition increased slightly with age. Scores of the reactive condition were the lowest, compared with those in the static and active conditions. 

The mean (SD) AIMS subscale and total scores are shown in Table 3. The subscale of each posture and the total scores of AIMS increased through the corrected ages of 4 and 5 months, and at the age of independent sitting attainment. Participants gained full AIMS scores in the supine subscale at the age of sitting onset.

There were significant correlations between the level of trunk segment control of all 3 conditions, the AIMS sitting subscale scores at 4 and 5 months corrected, and at the onset of independent sitting (Table 4). Significant correlations between the median level of trunk control in all conditions and the AIMS sitting subscale scores increased with age. Results showed significant fair correlations (r_s_ = 0.370–0.418, *p* < 0.05) at 4 months corrected age, fair-to-good correlations (r_s_ = 0.369–0.649, *p* < 0.05) at 5 months, and moderate-to-good (r_s_ = 0.561–0.602, *p* < 0.05) correlations at the onset of independent sitting.

## 4. Discussion

The current prospective study hypothesised that segmental postural control was correlated with sitting ability during the development of sitting milestones in home-raised moderate-to-late premature infants aged 4 and 5 months and at the onset of attained independent sitting. The results confirmed the hypothesis, demonstrating significant correlations between SATCo scores and the AIMS sitting subscale scores at each point of data collection. The significant correlations increased with age from fair to good. Corresponding to these results, the effect sizes from 4 months corrected age until the onset of independent sitting attainment also increased.

At 4 months corrected age, infants were not independent sitters, as they learned to control their trunk segments at the midthoracic in the static condition, and upper thoracic level in the active and reactive conditions, as shown by the median scores of the SATCo (Table 2). Their control of trunk segments progressively developed to the lower thoracic in the static condition, midthoracic level in the active and reactive conditions at 5 months corrected age, and to the full trunk in the static condition to the upper lumbar region in the active and reactive conditions at the age of independent sitting attainment. Since we defined the onset of sitting independently as the ability to momentarily sit without support, infants in our study did not show full control of trunk segments in the active and reactive conditions. 

Our study confirmed that the level of trunk control in segments progressively develops in the cephalocaudal direction with increasing age. All static, active, and reactive conditions of trunk control increased with age. The static condition of trunk control developed first, as shown by the highest score of reactive trunk control, compared with active and reactive trunk control. Although SATCo scores in the reactive condition were the lowest, the significant correlation between SATCo and AIMS scores was highest in the reactive condition, compared with that of other conditions. This happened from 4 months to the age of independent sitting onset, and this was increasing with age. The high correlation may imply the importance of reactive trunk control during sitting development in premature infants. In other words, the reactive condition could be the most important response to the degree of difficulty in controlling upright posture. In practice, infants who experience difficulties in trunk segment control show excessive sway and a wide range of motion while controlling the trunk in space during sitting. We assumed that the reactive condition of the SATCo test was sensitive in discriminating the ability of trunk segment control in sitting posture.

Correlations of the level of trunk control and the sitting ability suggest that trunk segment control is partly attributed to the ability to be in an upright position, such as independently sitting and standing. A cross-sectional study of Greco et al. in 2020 [12] verified the correlation between trunk control and the acquisition of gross motor performance in prone, supine, sitting, and standing positions at 6 and 7 months of age in 26 moderate-to-late preterm infants who had had an average gestational age of 33 weeks and were without comorbidity from prematurity. They found a significantly high correlation (r = 0.77; *p* value = 0.00) between the level of trunk segment control at the lower thoracic and sitting, measured by the AIMS at 7 months corrected age. The results of our study were in line with the results of this cross-sectional study. At the age of independent sitting, which was around 7 months, there were significant moderate-to-good correlations between the level of trunk segment control at the upper lumbar level in active (r_s_ = 0.602, *p* value < 0.01) and reactive (r_s_ = 0.561, *p* value < 0.01) conditions, and sitting performance [12].

However, our results were apparently different from those of a previous longitudinal study in 31 extremely preterm infants with less than 30 weeks gestational age [11]. We found a significant correlation between the SATCo and the AIMS since the age of 4 months corrected age, while Pin et al. showed no significant correlation between the SATCo and AIMS scores at this age. The explanation of the difference could be that infants from this study were moderate-to-late preterm. Moreover, the previous study analysed correlations of preterm and full-term infants together (*n* = 59). This previous study found significant correlations between trunk segment control and gross motor performance, such as prone, sitting, and standing at 8 and 12 months of age. It is predictable that preterm infants born at less than 32 weeks of gestation with biological risks could present delayed trunk-control development. Thus, the nonsignificant correlation at 4-month corrected age between trunk segment control and gross motor development was linked to having increased difficulties with maintaining an upright posture of sitting and standing at 4-month corrected age for very premature infants [11]. The results of our study show the development of trunk segment control from the static to the active, and lastly, the reactive condition, as shown by the highest score in the static condition at each age (Table 2). Significant correlations from 4-month corrected age until the corrected age of independent sitting could suggest that the developmental changes of trunk control in segments contribute to the ability to be upright, especially in sitting posture. This occurs in younger moderate-to-late preterm infants, even though they are not able to sit independently. Moreover, late premature infants in the current study showed onset of momentary sitting without support at the mean corrected age of 7 months, 9 days (SD, 1 month, 8 days), which is within the range of the window of development in typically developing children [1].

Longitudinal assessments of trunk segment control in our study confirmed the perspective of the instability of the gross motor skill found by Darrah et al. [20,21,22]. The range of trunk control at 4- and 5-month corrected age showed large variability from the head to the lower lumbar levels among the 3 conditions, while trunk segment control at the onset of independent sitting showed a narrower range from the lower thoracic to full trunk control. This result supports the previous study asserting that while infants had mastered new motor skills, they showed less variability of gross development via longitudinal assessments [20]. From this view, we accept that the increasing variability of postural control during sitting development reflects the experience of infants in maintaining postural control in each condition. Further study could consider a longer follow-up of trunk segment control until the full development of trunk control is obtained or infants become mature sitters.

## 5. Conclusions

The present study examined correlations between the control of trunk segments with sitting milestone development via a longitudinal study in moderate-to-late premature infants. We found a series of significant correlations from 4 months until the early onset of attaining independent sitting in moderate-to-late premature infants. This group of preterm infants did not demonstrate delayed independent sitting attainment. In addition, the SATCo is a valid and reliable test for assessing the development of trunk segment control. With the different challenging conditions of the test, it is practical for research and clinical practice. The SATCo could be considered to be a continuous monitoring assessment of sitting development in premature infants. If premature infants obtain continuous low scores over time, they may benefit from postural control stimulation provided by physiotherapists.

## Figures and Tables

**Figure 1 children-08-00722-f001:**
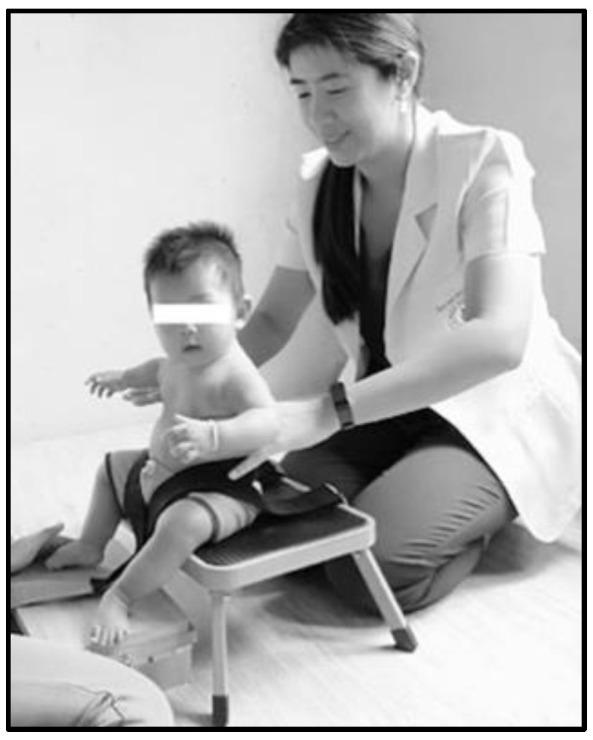
Segmental Assessment of Trunk Control (SATCo) includes 7 trunk segments.

**Table 1 children-08-00722-t001:** Demographic data of participants (*n* = 33).

Demographic Data	Mean (SD)	Range
Birth weight (g)	2139.5 (406.7)	1330–3092
Birth length (cm)	47.5 (1.7)	44–50
Birth head circumference (cm)	30.4 (1.5)	29–34
Apgar Score at 5 min	9.3 (0.5)	8–10
Average gestational age	34.5 (1.5)	32–36

**Table 2 children-08-00722-t002:** SATCo scores of each condition at 4- and 5-month corrected ages, and at the corrected age of independent sitting (*n* = 33).

Corrected Age of Prematurity	SATCo Conditions	Level of Trunk Segment	Median	Range
4 months	Static	Midthoracic	3	2 to 5
	Active	Upper thoracic	2	1 to 4
	Reactive	Upper thoracic	2	1 to 4
5 months	Static	Lower thoracic	4	3 to 6
	Active	Mid thoracic	3	2 to 5
	Reactive	Mid thoracic	3	2 to 4
At age of independent sitting	Static	Full trunk	7	5 to 7
	Active	Upper lumbar	5	4 to 7
	Reactive	Upper lumbar	5	4 to 6

Numbers are median and range of SATCo scores and also represent the level of trunk segment: 1 = head, 2 = upper thoracic, 3 = midthoracic, 4 = lower thoracic, 5 = upper lumbar, 6 = lower lumbar, 7 = full trunk control.

**Table 3 children-08-00722-t003:** Mean (SD) total and subscale of AIMS scores at 4- and 5-month corrected ages, and at the corrected age of independent sitting (*n* = 33).

AIMS	4 Months	5 Months	Age of Independent Sitting
Prone	5.1 (1.2)	7.6 (1.8)	12.9 (2.2)
Supine	5.9 (1.8)	7.6 (1.2)	9.0 (0.2)
Sitting	2.4 (0.6)	3.7 (1.0)	7.3 (0.7)
Standing	1.9 (0.3)	2.0 (0.0)	2.6 (1.4)
Total	15.3 (2.8)	21.0 (3.4)	31.7 (3.6)

**Table 4 children-08-00722-t004:** Spearman’s rho correlations between SATCo and AIMS sitting subscale scores at 4- and 5-month corrected ages, and at the corrected age of independent sitting (*n* = 33).

SATCo	Correlation Coefficients (r_s_)		
Conditions	4-Month Corrected Age	5-Month Corrected Age	Age of Independent Sitting
Static d (effect size)	0.370 *	0.369 *	0.561 **
	0.780	0.830	1.590
Active d (effect size)	0.420 *	0.400 *	0.561 **
	0.800	0.920	1.590
Reactive d (effect size)	0.418 *	0.649 **	0.602 **
	0.980	2.180	1.830

* Significant correlation (*p* < 0.05); ** significant correlation (*p* < 0.01).

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
