# Peer review of "Segmental Assessment of Trunk Control in Moderate-to-Late Preterm Infants Related to Sitting Development"

_children, 2021, doi:10.3390/children8090722_

Round 1

Reviewer 1 Report

Major comments:

  1. Introduction:
  • The aims should be rewritten: please provide the concise aims of the study. E.g. as bullet list or numbers
  • The definition of independent sitting belongs in the Methods section, not in the Introduction
  1. Figure 1

I don't understand the legend. What is meant by "segmental trunk"? Better: includes 7 trunk segments

  • M&M
  • The results of the ICC belong in the Results section
  • How were the assessments of the two assessors compared? Or did you use the mean scores of the two assessors? This needs to be explained
  1. Results:
  • I don’t understand this table: In the legend it is written that the numbers represent levels of trunk control or do these numbers represent SATCo scores? This needs more and better explanation.
  1. Conclusions:

Please comment on the clinical significance of the findings: E.g. if the SATCo scores are low should interventions such as physiotherapy be initiated?

Minor comments:

  1. Introduction
  • Line 36: sitting between 3.6-9.2 months. Isn’t that very early? Is this correct? Or are Asian children much faster than European children? Please explain
  • Line 72: in instead of between
  • Line 72: gestational age instead of “gestation at”
  • Line 76: the same group instead of they
  • Line 77: two studies (you are referring here to 11 and 12) instead of another previous study
  • Line 80: omit they found that
  • Line 82-85: I don't understand the last part of this sentence (after the comma). What do you want to say here?
  • Line 85: omit that
  1. M&M
  • Line 104: which University? Which hospitals?
  • Line 105: add moderate-late
  • Line 105: between instead of with
  • Line 107: what do you mean with uncontrolled seizure? Did you include infants with seizures that were under control with medication? Please explain. This does not agree with “healthy preterm infants”
  • Line 108 and 109: classification PVL and IVH according to? De Vries? Papile? Volpe?
  • Line 123: for instead of “that gives a specific”
  • Line 186-187: Initials of the two physiotherapists? Are these co-authors?
  • Data-analysis: how were the assessments of the two assessors compared (see par 2.4)?
  • Results
  • Omit First two sentences (line 213-217)
  • Line 218: what is meant by mean age of admission 4months + 5 days? Is this mean age of recruitment or mean each of first assessment? Please explain
  • Lines 220-222: They were not all healthy? 9 infants had BPD? See above. Please explain as in the M&M section it is written that healthy preterm infants were included
  • Table 2: corrected age of prematurity instead of preterm
  1. Discussion
  • Line 283: omit degree of
  • Line 337: sitting and standing at 4 mnths corrected age seems very young and not realistic? Please explain
  1. Conclusion
  • Line 365 add infants (preterm infants

Author Response

Responses to the Reviewers’ Comments

Thank you very much all reviewers for giving us an opportunity to make the revision of the manuscript entitled Segmental assessment of trunk control in moderate to late preterm infants related to sitting development. We have revised the manuscript and response to reviewers point by point as following.

Comments from Reviewer #1

Major comments:

  1. Introduction:

1) Reviewer comment:

The aims should be rewritten: please provide the concise aims of the study. E.g. as bullet list or numbers.

Response (page 2 / line 85-88)

We have rewritten the aims of the study according to the one in the abstract and put it in the new paragraph.

2) Reviewer comment:

The definition of independent sitting belongs in the Methods section, not in the Introduction

Response (page 4 / line 185-187)

The definition of independent sitting has been moved to the method section as suggested.

3) Reviewer comment:

Figure 1 I don't understand the legend. What is meant by "segmental trunk"? Better: includes 7 trunk segments

Response (page 3 / line 143)

We have changed "segmental trunk" to "trunk segments."

  1. Material and methods

4) Reviewer comment:

The results of the ICC belong in the Results section. How were the assessments of the two assessors compared? Or did you use the mean scores of the two assessors? This needs to be explained.

Response: (page 5 / line 206-209)

We have moved the results of the ICC to the results section. We have rewritten the procedure about the tests performing of the assessors. In the intra rater reliability, we compared the mean score of two assessments within the same rater. We did not compare the mean score of the two assessors because they performed different tests.

  1. Results

5) Reviewer comment:

I don’t understand this table: In the legend it is written that the numbers represent levels of trunk control or do these numbers represent SATCo scores? This needs more and better explanation

Response: (page 5 / Table 2)

We have put more explanation on the legend of the Table 2 and changed the heading of the third column to “level of trunk segment.”

The new legend was written as “Numbers are median and range of SATCo scores, and also represent the level of trunk segment: 1 = head, 2 = upper thoracic, 3 = midthoracic, 4 = lower thoracic, 5 = upper lumbar, 6 = lower lumbar, 7 = full trunk control”.

  1. Conclusions:

6) Reviewer comment:

Please comment on the clinical significance of the findings: E.g. if the SATCo scores are low should interventions such as physiotherapy be initiated?

Response: (page 8 / line 342-344)

This longitudinal study performed the SATCo test from the 4 months corrected age to the early onset of the sitting independently, we assumed that the trunk control in each segment still need to be develop to their maturity or when infants can steadily sit by themselves. Due to no cut off SATCo score for the preterm we cannot determine that infants from this study who got a low score will be delayed sitter. Therefore, we would recommend that a series assessment or longitudinal assessment of SATCo is needed in premature birth. “If premature infants get a continuous low score overtimes, then they need the postural control stimulation from physical therapy.”

We have added the recommendation in. conclusion as follow.

“The SATCo could be considered to be a continuous monitoring assessment of sitting development in premature infants. If premature infants obtain continuous low score overtimes, then they need the postural control stimulation from physical therapy.”

  1. Minor comments:

Introduction

7) Reviewer comment:

Line 36: sitting between 3.6-9.2 months. Isn’t that very early? Is this correct? Or are Asian children much faster than European children? Please explain.

Response:

The sitting onset between 3.8-9.2 months in full-term infants is referenced by a multicenter study of five countries (Brazil, Ghana, India, Norway, Oman and the USA.) by WHO (WHO, 2006). The range of the sitting onset could be widely varied according to child-rearing practice (Figure below, WHO 2006). The onset of sit independently is different in other cultures might vary from child-rearing practice. Most studies of sitting onset in infants around the world have various rates based on challenges of practice and contextual factors of specific cultures. Several researchers focused on climates and physical conditions affected by intra-individual and inter-individual development changes. The American infant shown sits independently at 6 months. In contrast, typical normative of Uganda infants were remarkably advanced in sit independently at 4 months (Adolph et al., 2010).

Adolph, K.; Karasik, L.; Tamis-LeMonda, C. Handbook of cross-cultural developmental science. Domains of development across cultures. New York 2010.

Modified from the WHO study (WHO, 2006)

WHO Multicentre Growth Reference Study Group. WHO Motor Development Study: windows of achievement for six gross motor development milestones. Acta Paediatr 2006, 95, 86-95, doi:https://doi.org/10.1111/j.1651-2227.2006.tb02379.

8) Reviewer comment:

Line 72: in instead of between

Line 72: gestational age instead of “gestation at”

Line 76: the same group instead of they

Line 77: two studies (you are referring here to 11 and 12) instead of another previous study

Line 80: omit they found that

Line 85: omit that

Response:

We have edited the above text as suggested.

On page 2 / line 72

On page 2 / line 72

On page 2 / line 75-76

On page 2 / line 77

On page 2 / line 80

On page 2 / line 83

9) Reviewer comment:

Line 82-85: I don't understand the last part of this sentence (after the comma). What do you want to say here?

Response: (page 2 / line 81-84)

We have revised the text as follows “There was limited longitudinal data on a relationship between the development of trunk segment control and sitting ability [12]. We hypothesised there would be relationships between the segmental trunk control and sitting ability from the age of learning to sit or at 4 months corrected age onwards.”

Material and methods

10) Reviewer comment:

Line 104: which University? Which hospitals?

Line 105: add moderate-late

Line 105: between instead of with

Response:

We have edited the above text as suggested.
On page 2 / line 96-97

On page 2 / line 98

On page 3 / line 99

11) Reviewer comment:

Line 107: what do you mean with uncontrolled seizure? Did you include infants with seizures that were under control with medication? Please explain. This does not agree with “healthy preterm infants”

Response: (page 3 / line 100)

We recruited healthy preterm infants with no history of seizure, so we have changed the text as followed.

“The exclusion criteria were infants with known seizure.”

12) Reviewer comment:

Line 108 and 109: classification PVL and IVH according to? De Vries? Papile? Volpe?

Response: (page 3 / line 102)

We have put the referenced in the revised manuscript.

The PVL classification, we have used the reference as follows

de Vries, L.S.; Eken, P.; Dubowitz, L.M. The spectrum of leukomalacia using cranial ultrasound. Behavioural brain research 1992, 49, 1-6, doi:https://doi.org/10.1016/S0166-4328(05)80189-5.

The IVH classification, we have used the reference as follows

Papile, L.-A.; Burstein, J.; Burstein, R.; Koffler, H. Incidence and evolution of subependymal and intraventricular hemorrhage: a study of infants with birth weights less than 1,500 gm. The Journal of pediatrics 1978, 92, 529-534, doi:https://doi.org/10.1016/S0022-3476(78)80282-0.

13) Reviewer comment:

Line 123: for instead of “that gives a specific”

Response:

We have edited the above text as suggested.

On page 3 / line 116

14) Reviewer comment:

Line 186-187: Initials of the two physiotherapists? Are these co-authors?

Response

N.S is the first author and R.T is the research assistant.

15) Reviewer comment:

Data-analysis: how were the assessments of the two assessors compared (see par 2.4)?

Response: (page 4 / line 180-183)

We did not compare the mean score of the two assessors because they performed different tests. But we analyzed the mean score of each assessor twice using the ICC (model 3,1).

Results

16) Reviewer comment:

Omit First two sentences (line 213-217)

Response:

We have edited the above text as suggested.

On page 4 / line 201

17) Reviewer comment:

Line 218: what is meant by mean age of admission 4months + 5 days? Is this mean age of recruitment or mean each of first assessment? Please explain

Response

4 months and 5 days was the mean age of the first assessment. We intended to collect the first data at 4 months, but some infants were not ready, so we allowed approximately 5 days after 4 months to collect the data.

18) Reviewer comment:

Lines 220-222: They were not all healthy? 9 infants had BPD? See above. Please explain as in the M&M section it is written that healthy preterm infants were included

Response

According to our data, these 9 infants had a history of mild BPD after birth, but at the first data collection they were healthy and met the inclusion criteria of the study.

19) Reviewer comment:

Table 2: corrected age of prematurity instead of preterm

Response:

We have edited the above text as suggested.

On page 5 / Table 2

Discussion

20) Reviewer comment:

Line 283: omit degree of

Response:

We have edited the above text as suggested.

On page 7 / line 273

21) Reviewer comment:

Line 337: sitting and standing at 4 moths corrected age seems very young and not realistic? Please explain.

Response

“Non-significant correlation at 4 months corrected age between segmental trunk control and gross motor development has been linked to having increased difficulties with maintaining an upright posture of sitting and standing at 4 months corrected age for very premature infants.”

The above text means a previous study (Pin et al., 2020) found no correlation between trunk control and gross motor at 4 months. They suggested that very premature infants at this age show no trunk control in the segment as they have difficulty maintaining an upright posture.

Conclusion

22) Reviewer comment:

Line 365 add infants (preterm infants).

Response:

We have edited the above text as suggested.

On page 8 / line 338

Reviewer 2 Report

Thank you for allowing me to review the article titled "Segmental assessment of trunk control in moderate to late preterm infants related to sitting development" by Sangkarit et al.

My comments for minor changes:

What is the purpose of using corrected age?

Define moderate to late preterm? it is better to use mid to late preterm

Author Response

Responses to the Reviewers’ Comments

Thank you very much all reviewers for giving us an opportunity to make the revision of the manuscript entitled Segmental assessment of trunk control in moderate to late preterm infants related to sitting development. We have revised the manuscript and response to reviewers point by point as follows.

Comments from Reviewer #2

1) Reviewer comment:

What is the purpose of using corrected age?

Response

Preterm birth defined by gestational age can be classified into 3 categories as extremely preterm (less than 280/7th weeks), very preterm (between 280/7th to 316/7th weeks), moderate to late preterm (between 320/7th to 366/7th weeks). The category reflects marked differences in the probability of survival, intensive care costs, long-term health status and disability outcomes (Engle, 2004; Lumley, 2003; McGowan et al., 2011).

Engle, W.A. Age terminology during the perinatal period. Pediatrics 2004, 114, 1362-1364.

Lumley, J. Defining the problem: the epidemiology of preterm birth. BJOG: An International Journal of Obstetrics & Gynaecology 2003, 110, 3-7.

McGowan, J.E.; Alderdice, F.A.; Holmes, V.A.; Johnston, L. Early childhood development of late-preterm infants: a systematic review. Pediatrics 2011, 127, 1111-1124.

In general, any research which has done in premature birth used corrected age in order to be compared with previous studies or with full-term infants. Therefore, we used corrected age.

2) Reviewer comment:

Define moderate to late preterm? it is better to use mid to late preterm.

Response

In order to be consistent with terminology of literature in this field we used moderate to late preterm according to (WHO, 2018) [1]

World Health Organization WHO. Preterm birth. 2018, doi:https://www.who.int/news-room/fact-sheets/detail/preterm-birth.

Round 2

Reviewer 1 Report

See attached file

Author Response

Responses to the Reviewers’ Comments

Thank you very much for suggestion in minor revision. We have revised the manuscript and response to reviewers point by point as following.

Comments from Reviewer #1

Comments:

1) Reviewer comment:

Line 88: this study examined segmental trunk control (instead of segmentally

examined)

Response

We have edited the above text as suggested.
On page 2 / line 88

2) Reviewer comment:

Line 121: Exclusion criteria were infants with known seizures

Response (page 3 / line 101)

The words have been removed as suggested.

3) Reviewer comment:

Line 209 and 211: pediatric physical therapist

Response (page 4 / line 189 and 191)

We have changed from to "therapy" to "therapist" as suggested.

4) Reviewer comment:

Line 209 and 211: please proved the initials of these examiners (if they are co-

authors)

Response

N.S. is Miss Noppharath Sangkarit who performed the SATCo

R.T. is Miss Rungrudee Tupsila who performed the AIMS and is not the co-author of the current study.

page 4 /line 190 and 191

5) Reviewer comment:

Line 212: and the scores of the two separate examiners were blinded to each other.

Response: (page 4 / line 192)

We have added "two separate examiners" as suggested.

6) Reviewer comment:

Results lines 230-251: this belongs to the M&M section (in par 2.4). Therfore the

Results start with: The intrarater reliability of the SATCo.

Response

The word has been moved to the method section as suggested.

On page 4 / line 182-188

7) Reviewer comment:

Line 260: again, these infants were not completely healthy. This needs some

clarification. Maybe you should add this to M&M par 2.1 (Participants): explain that

the infants were healthy at inclusion but not necessarily during the neonatal period.

Response

We have removed the word “healthy” from the manuscripts and changed the inclusion criteria to “Moderate-to-late premature infants born between 32–36 weeks of gestation were recruited at 4 months corrected age for the first data collection if infants had stable health condition.”

On page 2 / line 99 - page 3 line 104-105

8) Reviewer comment:

Line 313: omit (Table 4)

Response:

We have omitted the above text as suggested.

On page 6 / line 263

9) Reviewer comment:

Line 314: add (Table 4) at the end of the sentence.

Response

We have added "(Table 4)" at the end of the sentence as suggested.

(page 6 / line 264)

10) Reviewer comment:

Line 324: moderate-late preterm infants

Response:

We have edited the above text as suggested.
On page 6 / line 280

11) Reviewer comment:

Line 345: add between SATCo and AIMS scores (the correlation between SATCo and

AIMS scores....)

Response

We have added the above text as suggested.

On page 7 / line 296-297

12) Reviewer comment:

Lines 346-349: or is it the other way around? One needs trunk control to reach

independent sitting? Please add some comment on this, also in relation to the next

sentences.

Response

We have revised the text as follow “This happened from 4 months to the age of independent sitting onset, and this was increasing with age. The high correlation may imply the importance of reactive trunk control during sitting development in premature infant. In other words, the reactive condition could be the most important response to the degree of difficulty in controlling upright posture.”

On page 7 / line 305-309

13) Reviewer comment:

Line 361: they found a significant high correlation.

Response:

We have edited the above text as suggested.

On page 7 / line 320

14) Reviewer comment:

Line 370: extreme instead of extremely.

Response

We have edited the above text as suggested.

On page 7 / line 329

15) Reviewer comment:

Lines 370-377: the fair correlations mentioned here are a bit confusing to me. Can’t

you just write: significant correlations?

Response

We have edited the above text as suggested.

On page 7 / line 330 and 334

16) Reviewer comment:

Line 386: could suggest instead of “could be suggesting”

Response:

We have edited the above text as suggested.

On page 7 / line 344-345

17) Reviewer comment:

Line 389: sit independently instead of “independently sit”

Response

We have edited the above text as suggested.

On page 7 / line 347

18) Reviewer comment:

Line 416: they may benefit from postural control stimulation provided by

physiotherapists instead of “they then need....”

Response

We have edited the above text as suggested.

On page 8 / line 394-395
